# Amorphous Calcium Carbonate from Plants Can Promote Bone Growth in Growing Rats

**DOI:** 10.3390/biology13030201

**Published:** 2024-03-21

**Authors:** Chun-Kai Chen, Yu-Shan Lee, Zwe-Ling Kong, Yi-Wen Chien

**Affiliations:** 1Department of Food Science, National Taiwan Ocean University, Keelung 202301, Taiwan; dennis.ckchen@gmail.com; 2Department of Nutrition and Health Science, Taipei Medical University, Taipei 11031, Taiwan; ma07111010@tmu.edu.tw; 3Research Center of Geriatric Nutrition, College of Nutrition, Taipei Medical University, Taipei 11031, Taiwan; 4Graduate Institute of Metabolism and Obesity Sciences, Taipei Medical University, Taipei 11031, Taiwan; 5Nutrition Research Center, Taipei Medical University Hospital, Taipei 110301, Taiwan; 6TMU Research Center for Digestive Medicine, Taipei Medical University, Taipei 110301, Taiwan

**Keywords:** growing rat, calcium, supplementation, bone mineral density, bone turnover marker

## Abstract

**Simple Summary:**

In order to explore the effect of amorphous calcium carbonate (ACC) supplementation on bone growth in growing rats, 3-week-old male Wistar rats were divided into four groups as follows: a control group (C), low-dose group (L, 20.65 mg/kg of body weight (BW) ACC), a medium-dose group (M, 206.5 mg/kg of BW ACC), and a high-dose group (H, 413 mg/kg of BW ACC) administered by gavage for 12 weeks. The results indicated that ACC supplementation can enhance osteoblast metabolism and inhibit osteoclast metabolism, resulting in a higher bone formation rate compared to bone resorption. This led to increased trabecular bone thickness and a higher bone mineral density (BMD) and supported bone growth. By increasing peak bone mass, it is speculated that the risk of future osteoporosis could be reduced. To build upon the positive results of our study on skeletal health, future follow-up studies could be conducted on older rodents treated with ACC supplementation during the growth period.

**Abstract:**

Objectives: This study aimed to investigate the effect of amorphous calcium carbonate (ACC) supplementation on bone growth in growing rats. Methods: We used 3-week-old male Wistar rats to simulate childhood and adolescent growth stages. Rats were divided into four groups as follows: a control group (C), a low-dose group (L, 20.65 mg/kg body weight (BW) ACC), a medium-dose group (M, 206.5 mg/kg BW ACC), and a high-dose group (H, 413 mg/kg BW ACC) administered by gavage. Body length (BL) and BW were measured weekly. The bone mineral density (BMD) of two lumbar vertebrae (L3 and L4) and the left femur were analyzed by micro-computed tomography (μCT) at 0, 4, 8, and 12 weeks. At the end of 12 weeks, the rats were sacrificed. After that, blood samples were collected from the abdominal aorta. Femurs and tibias were collected and weighed, and their lengths were measured. Then, bone samples were used to perform histopathological and histomorphometric analyses. Results: It showed that ACC supplementation in growing rats increased the trabecular bone thickness and serum bone formation biomarkers. Furthermore, high-dose ACC decreased serum bone resorption biomarkers and increased BMD. Conclusions: ACC supplementation can enhance osteoblast metabolism and inhibit osteoclast metabolism, resulting in a higher bone formation rate compared to bone resorption. This led to increased trabecular bone thickness, a higher BMD, and supported bone growth.

## 1. Introduction

The World Health Organization (WHO) emphasizes that osteoporosis is the second most prevalent disease worldwide, following coronary heart disease. According to the 2007 “Taiwan Longitudinal Study in Aging” (TLSA) conducted by the Health Promotion Administration (HPA), osteoporosis is a common chronic condition among women over the age of 65 years. Furthermore, the “Nutrition and Health Survey in Taiwan” (NAHSIT) conducted from 2017 to 2020 revealed that approximately one in seven individuals aged 65 years and older had osteoporosis. Osteoporosis significantly increases the risk of fractures, which can lead to disabilities, higher hospitalization rates, reduced self-care ability, and even fatalities. It poses a substantial burden on both individuals and society. Therefore, early detection, intervention, and fracture prevention are crucial priorities [1].

Calcium is an essential mineral for bone development and maintaining bone health. Calcium plays a key role in inhibiting bone loss and is a vital component in bone synthesis. Bone mass gradually increases from childhood, and rapid growth is experienced during adolescence. With proper nutrition and weight training, this increase in bone mass can continue to the age of 35 to 40 years when peak bone mass (PBM) is reached. Approximately 60% of osteoporosis cases in later life result from inadequate PBM accumulation during the growth period. Increasing PBM by 10% during the growth period can reduce the risk of fractures related to osteoporosis by 50% [2,3].

According to the 2017–2020 NAHSIT conducted by the HPA, calcium has low intake among Taiwanese individuals. Calcium intake among Taiwanese individuals over 4 years of age falls below the recommended level. As a result, calcium supplements have become a significant source of calcium in modern life. Calcium supplements are available in various forms, with calcium citrate and calcium carbonate being the most common. Calcium carbonate can be further categorized into crystalline calcium carbonate (CCC) and ACC. Among these, CCC calcite is the most thermodynamically stable, while ACC is the least stable [4,5].

ACC is a transient polymorph containing 33% elemental calcium, formed by precipitation from a supersaturated solution following Ostwald’s step rule. Unless stabilized by other elements, it quickly and entirely crystallizes into other forms of calcium carbonate [6]. In comparison to CCC, ACC consists of smaller particles (CCC: 1~10 μm; ACC: 40~120 nm) and exhibits greater solubility and bioavailability [4]. Compared with CCC and calcium citrate, ACC has advantages in preventing bone loss, promoting bone formation, and enhancing bone strength [7]. Therefore, ACC may represent a more efficient form of calcium supplementation.

There have only been a limited number of studies investigating the relationship between ACC and bone mass, with the majority focusing on postmenopausal conditions. One study reported that the daily administration of a standard calcium diet (5.3 g Ca/kg diet) using ACC as the calcium source for ovariectomized rats over 90 days of age increased the amount of trabecular bone in the femur, tibia, and spine compared to the CCC group. Additionally, it increased the trabecular bone thickness of the tibia and reduced deoxypyridinoline (DPD) concentrations in the blood [7]. Another study found that when 600 mg of ACC was administered to postmenopausal women (equivalent to 192 mg of calcium), ACC had greater bioavailability compared to the same dose of CCC [8]. These findings suggest that ACC might have a positive impact on bone development during growth.

As a consequence, our study aimed to investigate the effects of ACC supplementation on bone growth in growing rats in order to understand whether supplementation with ACC in children and adolescents in the growing stage can promote bone growth.

## 2. Materials and Methods

### 2.1. Animals

In total, 33 male Wistar rats at 3 weeks old were purchased from BioLasco (Taipei, Taiwan). All animals were housed in an animal room with a 12 h light/dark cycle (with lights on at 07:00). In addition, the temperature was controlled to 22 ± 2 °C, and the relative humidity was 65% ± 5%. The experiment began after 1 week of acclimatization. During the experiment, the AIN-93G diet was administered, and rats were free to eat and drink ad libitum. This experiment was approved by the Institutional Animal Care and Use Committee of Taipei Medical University (IACUC no.: LAC-2022-0265).

### 2.2. Experimental Design

All rats were randomly assigned to different experimental groups, including a control group (*n* = 9) (C, which received 10 mL of double-distilled (dd)H_2_O/kg BW/day), a low-dose group (*n* = 8) (L, 20.65 mg of ACC/kg BW/day), a medium-dose group (*n* = 8) (M, 206.5 mg of ACC/kg BW/day), and a high-dose group (*n* = 8) (H, 413 mg ACC/kg BW/day). ACC (Amorphical, Beer-Sheva, Israel) was mixed in ddH_2_O and given via gavage. During the 12 weeks of ACC intervention, BW, BL (with and without the tail), and food intake were recorded weekly. The food efficiency ratio (FER, %; food intake/BW gain) and Lee’s index (the cube root of body weight (g)/nose-to-anus length (cm)) were calculated. The BMD of two lumbar vertebrae (L3 and L4) and the left femur were measured by micro-computed tomography (μCT) at 0, 4, 8, and 12 weeks during the experiment. Rats were sacrificed with a mixed solution of zoletil and rompun (1 mL/kg BW, by intraperitoneal injection) at the end of 12 weeks of the experiment. The serum was collected and centrifuged at 3500× *g* and 4 °C for 15 min. The femurs and tibias were dissected and weighed, and their lengths were measured. All samples were stored at −80 °C until they were examined.

### 2.3. Bone Turnover Biochemical Markers (BTMs)

Serum bone alkaline phosphatase (BALP) levels were measured using an enzyme-linked immunosorbent assay (ELISA) kit (Wuhan Fine Biotech, Wuhan, China). Serum procollagen-I C-terminal propeptide (PICP) levels were measured with an ELISA kit (MyBioSource, San Diego, CA, USA). Serum cross-linked N-telopeptide (NTx) levels were measured with an ELISA kit (Chondrex, Redmond, WA, USA). Serum pyridinoline (Pyd) levels were measured with an ELISA kit (AFG Bioscience, Northbrook, IL, USA).

### 2.4. Micro-Computed Tomography (μCT)

The images of two lumbar vertebrae (L3 and L4) and the left femur were scanned by μ-CT (Skysacn, 1176, Kontich, Belgium) at 0, 4, 8, and 12 weeks during the experiment. We used a gas anesthesia machine to vaporize isoflurane (by heating it to about 40~50 °C). Rats were anesthetized with 2.0~2.5% isoflurane dissolved in oxygen. At this concentration, anesthesia can be completed within 5 to 10 min. After that, rats were positioned on a scan bed. At the same time, rats were continuously administered 1.5~1.8% of isoflurane by inhalation using an animal respirator to maintain the anesthetic state. All scans were measured with a nominal voxel size of 18 μm. Scanning parameters were set to 80 kV (X-ray tube), with an intensity of 250 μA and an exposure time of 740 ms. We calibrated BMD by following the manufacturer’s instructions utilizing specified rat phantom rods with known densities of 0.25 and 0.75 g/cm^3^ as the reference standards (Skyscan). We used NRecon (vers. 1.7.5.4) to reconstruct the scans, and the reconstructed images were reoriented by DataViewer (vers. 1.5.1). The BMDs of lumbar vertebrae (L3 and L4) and the left femur, the left femur trabecular bone volume (mm^3^), and left femur cortical bone thickness (μm) were calculated with a CT-Analyzer.

### 2.5. Bone Length and Weight

After rats were euthanized, the tibias and femurs were excised and cleared of adjacent soft tissues. We measured their weights, and the lengths were assessed with digital microcalipers (Digimatic Caliper, Mitutoyo, Japan).

### 2.6. Histopathological Examination

Following gross necropsy, the femurs were preserved in 10% neutral buffered formalin (NBF) and decalcified, trimmed, embedded in paraffin, sectioned, stained with hematoxylin and eosin (H&E), and examined microscopically by the study pathologist for trabecular bone amounts and morphological changes. The microstructure of the bone of the femur was histomorphologically evaluated after staining with H&E. The histomorphometric measurements of the trabecular thickness (μm) and trabecular area (%) were analyzed using ImageJ analytical software (vers. 1.54d).

### 2.7. Statistical Analysis

All data are presented as the mean ± standard deviation (SD). Statistical analysis was performed using the Statistical Package for the Social Sciences (SPSS, vers. 19.0, IBM, Armonk, NY, USA) software. Figures were performed using GraphPad Prism vers. 9.0 software (GraphPad Software, San Diego, CA, USA). First, we used the Shapiro–Wilk test to confirm whether the data were normally distributed. Then, A one-way analysis of variance (ANOVA) was used to compare the data among the four groups. When a statistically significant difference was observed, we conducted Duncan’s multiple-range test. Differences were regarded as statistically significant at *p* < 0.05.

## 3. Results

### 3.1. Body Weight (BW)

Figure 1 presents the BW changes in animals in each group at 0 to 12 weeks. The increase in BW was more rapid from 0 to 6 weeks and then gradually slowed down from 7 to 12 weeks. There were no significant differences in BWs among the four groups at 0 to 12 weeks. After 12 weeks of the ACC intervention, no differences were found in the total BW gain among the four groups (Table 1).

### 3.2. Body Length (BL)

Figure 2A presents the BLs of animals in each group without their tails from 0 to 12 weeks. Figure 2B presents the BLs of animals, including their tails, for the same duration. The increase in BL, whether with or without tails, was relatively steep from 0 to 4 weeks and gradually slowed down from 4 to 12 weeks. There were no significant differences in BLs, with or without tails, from 0 to 12 weeks among the four groups. After 12 weeks of ACC intervention, no differences were found in the total BL gain, with or without tails, among the four groups (Table 1).

### 3.3. Lee’s Index (LI)

The results in Table 2 present the LI for each group over the course of the 12-week experiment. In the 11th week, the LI of the H group was significantly lower than that of the L group (*p* < 0.05). No significant differences were observed at other time points.

### 3.4. Food Intake

There were no significant differences in food intake or food efficiency among the four groups (Table 1).

### 3.5. Bone Turnover Biochemical Markers (BTMs)

Figure 3 presents serum BTM levels after 12 weeks of the ACC intervention. The serum BALP level in the H group was slightly higher than that in the C group (*p* < 0.1). The serum PICP level in the H group was significantly higher than those in the L and C groups (both *p* < 0.05), while that of the M group was also significantly higher than that of the C group (*p* < 0.05). The serum Pyd level in the H group was significantly lower than that in the C group (*p* < 0.05). There were no significant differences in serum NTx levels among the four groups.

### 3.6. Bone Mineral Density (BMD)

The results in Figure 4 show changes in the BMD at each measurement point (for L3, L4, and the left femur) during the intervention in the four groups. At the baseline, there were no significant differences in BMD among the four groups. During the 4th week, there were no significant differences in the BMD of L3 and L4 among the four groups. However, the BMDs of the left femur in the three intervention groups (L, M, and H) were significantly higher than that in the C group (all *p* < 0.05). Additionally, the BMD of the left femur in the H group was significantly higher than that in the M group (*p* < 0.05). During the 8th week, there were no significant differences in the BMDs of L3 and L4 among the four groups. However, the BMDs of the left femur in the three intervention groups (L, M, and H) were significantly higher than that in the C group (all *p* < 0.05). During the 12th week, the BMD of L3 in the H group was slightly higher than that in the C group (*p* < 0.1), and the BMD of L4 in the M group was also slightly higher than that in the C group (*p* < 0.1). Otherwise, the BMD of the left femur in the H group was significantly higher than that in the C group (*p* < 0.05).

### 3.7. Bone Length and Weight

Figure 5 and Figure 6 present bone length and bone weight after 12 weeks of ACC intervention, respectively. There were no significant differences in the lengths or relative weights of the left and right femurs and tibias among the four groups.

### 3.8. Cortical Bone Thickness

Figure 7 presents cortical bone thicknesses on the 0, 4th, 8th, and 12th weeks during the experimental period. We observed an increase in cortical bone thickness over time. There were no significant differences in cortical bone thicknesses among the four groups at any of the time points (0, 4th, 8th, and 12th weeks).

### 3.9. Trabecular Bone

Figure 8 presents the histomorphometric analytical results of the left femur after 12 weeks of the ACC intervention. Trabecular thicknesses in the three intervention groups (L, M, and H) were significantly higher than those in the C group (*p* < 0.05). There were no significant differences in the trabecular area or trabecular volume among the four groups (Figure 9).

## 4. Discussion

Calcium is the most abundant mineral in the human body, with 99% present in the bones and teeth in the form of hydroxyapatite crystalline salts, while the remaining 1% is distributed in tissues and body fluids. Calcium is a primary component of the bones and teeth and plays essential roles in the functions of muscles, the cardiovascular system, the endocrine system, and the nervous system. It is an indispensable substance in the body. Adequate calcium intake during growth can help ensure healthy bones. However, it is important to note that adequate calcium intake is limited to supporting normal bone growth and development, and it does not cause an individual to grow taller than their predetermined genetic height. Height is primarily determined by genetic factors but can also be influenced by hormones, environmental factors, and nutrition. Therefore, while calcium is crucial for bone health and growth, it is not the sole factor determining a person’s height [9,10,11,12]. One study indicated that when growing rats were provided with a high-calcium diet (12 g Ca/kg diet) and a normal calcium diet (5 g Ca/kg diet) for 30 days, there were no significant differences in BL (without tails) between the two groups [13]. This aligns with the results of our experiment. Therefore, it can be concluded that this calcium tablet intervention experiment had no significant impact on BL, whether the rate was with or without tails.

Lee’s index (LI), also known as the Ponderal Index, is a metric used for assessing body size and proportions in smaller animals, such as rodents. Similar to the body mass index (BMI) in humans, it allows observations of changes in body size and proportions as animals grow. However, like the BMI, LI is not a comprehensive indicator of health. While it can provide insights into an animal’s growth, body composition, and potential health risks, it does not account for factors such as muscle mass, bone density, or body fat distribution. Therefore, it should be used in conjunction with other information for a more comprehensive assessment [14]. Significant differences in LI were observed during the 11th week, with group H having a significantly lower LI than group L (*p* < 0.05). No significant differences were found at other time points. This discrepancy might have been due to potential errors in measuring the length of living rats from 0 to 11 weeks. As evident in Figure 2, the BLs of the M and H groups during the 11th week exceeded the BL measured at the time of sacrifice in the 12th week. To reduce errors in future measurements, it is recommended to use gas anesthesia with isoflurane when measuring BLs.

During the growth and development stage, the body is particularly sensitive to factors that influence future health, such as BMD and peak bone mass (PBM). As a result, bone length and weight are not as significantly affected as BMD [15]. Previous studies have shown that calcium deficiency does not impact bone length in growing rats [16]. Similarly, providing growing rats with either a normal calcium diet (5 g Ca/kg diet) or a high-calcium diet (12 g Ca/kg diet) for 30 days did not affect their height or bone length [13]. Furthermore, some studies suggested that, regardless of whether growing rats were given a high- (10 g Ca/kg diet) or a low-calcium diet (2.5 g Ca/kg diet) for 10 weeks, bone length tended to be longer in heavier rats. When there was no significant difference in BW, there was no significant difference in the bone length of rats, whether they were given a high- or low-calcium diet. This indicates that bone length is related to BW [17]. Therefore, it is speculated that the absence of significant differences in bone length and weight among the groups in this experiment can be attributed to the following two factors: (1) during the growth and development process, variations in calcium intake have lower impacts on bone length and weight but are more likely to affect BMD and BTM; (2) since there were no significant differences in BW among the four groups, there were no significant differences in bone length.

The shape and structure of trabecular bone play crucial roles in bone quality and stability. A greater trabecular bone thickness indicates a more maturely formed trabecular bone. When the rate of bone loss surpasses the rate of bone synthesis, trabecular thickness decreases. The trabecular bone area is inversely related to the risk of osteoporosis and fractures, while trabecular bone volume reflects the overall bone mass [18,19,20]. Previous studies have shown that providing growing rats with adequate dietary calcium (5 g Ca/kg diet) daily for 6 weeks significantly increased trabecular bone thickness, regardless of whether the calcium source was calcium carbonate or *Opuntia ficus-indica* cladodes [21]. Another study demonstrated that providing growing rats with a high-calcium diet (12 g Ca/kg diet) daily for 30 days significantly increased both trabecular bone thickness and volume compared to a low-calcium group (2 g Ca/kg diet) [13]. The increased trabecular bone thickness aligns with the results of this experiment, indicating that the ACC intervention effectively increased the trabecular bone thickness. Although no significant differences were observed in the trabecular bone area or volume, the data suggested an upward trend from the low to the high dose. With an extended intervention time, more significant effects may have become evident.

BALP is a glycoprotein found on the cell membranes of bone cells, particularly osteoblasts. During the process of bone formation, osteoblasts secrete BALP. Subsequently, due to the action of phospholipase, it is released into the bloodstream, where it can interact with various metabolic pathways [22]. Previous research established that BALP serves as a valuable biochemical marker for assessing bone health [23]. Furthermore, since the BALP concentration tends to increase with the growth rate, it can be utilized as an indicator of the bone formation rate, especially during childhood and adolescence [24,25]. Type I collagen constitutes over 90% of the bone matrix. Enzymes that break down peptides cleave excess “carboxyl-terminal” peptides from procollagen to synthesize type I collagen. The rate of production of these removed PICP peptides is similar to the rate of mature type I collagen synthesis, making it a reflection of osteoblast activity. PICP is highly stable in the serum, making its serum concentration a valuable indicator of osteoblast metabolism [26]. Furthermore, the PICP concentration in the blood is positively correlated with the BALP concentration, so the simultaneous measurement of BALP and PICP can provide more accurate experimental results [24]. PYD is a cross-linked collagen peptide protein primarily found in the cartilage, bones, ligaments, and blood vessels. Its main function is to stabilize the formed collagen and enhance bone resorption by osteoclasts. During this process, collagen is broken down, releasing PYD into the bloodstream, where it can serve as an indicator of the rate of bone loss. As such, PYD concentrations in the blood are a valuable metabolic marker of bone loss. The PYD concentration is associated with fracture risks and exhibits both positive and negative correlations with bone density [21,26,27]. Type I collagen constitutes the primary component of the bone matrix, making up over 90% of it. During bone resorption by osteoclasts, collagenase enzymes break down type I collagen, releasing NTx into the bloodstream. NTx concentrations in the blood reflect the rate of bone loss. Higher NTx concentrations indicate faster bone loss rates. Consequently, measuring NTx in the blood can serve as a valuable metabolic indicator of bone loss [21,28,29].

In this experiment, we used PICP and BALP to measure bone synthesis (osteoblast activity) and PYD and NTx for bone loss (osteoclast activity). The human body constantly undergoes bone remodeling, which involves both bone creation and loss to maintain bone health despite daily wear and aging. This cycle comprises the following five stages: activation, bone loss, reversal, bone formation and mineralization, and termination. It starts with osteoclast activation, aided by cytokines at the remodeling site. These osteoclasts create an acidic environment on the bone tissue, breaking down the inorganic part. After that, enzymes further break down organic matter, forming small cavities. Once bone loss concludes, the process shifts to the reversal stage, marked by bone erosion. During this phase, cells transform into osteoblasts, which generate new bone. This continues until the bone surface is restored, and mature osteoblasts undergo apoptosis or become osteocytes. When bone synthesis surpasses bone loss, bone mass increases, but if bone synthesis lags behind, bone mass decreases [30,31,32,33]. In our experiment, we observed that PICP significantly increased in the intervention groups compared to the C group. BALP also showed a slight increase in the H group. This indicated that the intervention promoted bone synthesis. Additionally, the H group exhibited a significantly reduced PYD concentration compared to the C group, while no significant differences were found in NTx concentrations among the four groups. These results may have been due to using growing rats in the experiment. Overall, our intervention enhanced bone synthesis more than it reduced bone loss, leading to increased bone mass. In our study, we used 3-week-old male Wistar rats to simulate childhood and adolescent growth stages. The results of our study indicate that although supplementing ACC during the growth period has positive effects on bone mineral density (BMD), bone turnover markers (BTMs) and trabecular thickness, no relevant follow-up studies were conducted on elderly rats. However, osteoporosis primarily affects the elderly population. From the existing results, we can only infer that supplementing ACC during the growth period can improve bone health and potentially reduce the risk of osteoporosis in the future. Nonetheless, we cannot ascertain the impact this has on bone health at an older age; thus, this aspect is a limitation of our study. One study observed that ovariectomized rats aged 16–17 weeks were administered the same calcium content of ACC, CCC, and calcium citrate for 90 days (approximately 13 weeks). The trabecular bone mineral density of the femur and lumbar vertebrae in the ACC group was significantly higher than that of the CCC group and the calcium citrate group. Additionally, the bone resorption biomarker, Dihydropyrimidine (DPD), was significantly lower in the ACC group compared to the CCC group and the calcium citrate group. These findings suggest that ACC is more effective than the two most common calcium supplements (CCC and calcium citrate) in preventing bone loss [7]. To build upon the positive findings regarding skeletal health, previous studies have demonstrated the excellent performance of ACC in preventing bone loss. In the future, the duration of feeding could be extended, and follow-up studies could be conducted on elderly rodents treated with ACC supplements during the growth period. It is anticipated that, compared to the control group not receiving calcium supplements, better bone status will be exhibited (e.g., BMD), thereby potentially reducing the incidence of osteoporosis in the elderly.

Previous studies demonstrated that providing 1000 mg Ca/day to adolescent females for 7 years and 300 mg Ca/day to children for 1.5 years significantly increased BMD during growth [34,35]. In our study, growing rats were given a high-calcium diet (12.5 g of Ca/kg diet) or a normal diet (5 g of Ca/kg diet) every day for 9 weeks. The high-calcium group exhibited a significant increase in BMD compared to the low-calcium group (2.5 g of Ca/kg diet) [16]. In a separate experiment, growing rats were fed a high-calcium diet (12 g of Ca/kg diet) or a normal diet (5 g of Ca/kg diet) daily for 30 days. The high-calcium group also showed higher BMD compared to the low-calcium group (2 g of Ca/kg diet) [13]. These findings align with the results of our experiment, suggesting that calcium supplementation has the potential to increase BMD in growing male rats.

Cortical bone, or compact bone, forms the outer layer of most bones. It consists of tightly packed lamellae and osteocytes and is known for its strength, with the ability to withstand pressure, tension, and torsion. This makes cortical bone thickness a useful indicator for assessing local bone health [36,37]. Figure 7 displays the changes in left femoral cortical bone thickness before and after the experiment in each group. The results reveal an increase in cortical bone thickness over time, with no significant variations among the four groups. It is important to note that factors like exercise, weight, and height, not just calcium intake, influence cortical bone thickness [38]. A study involving growing rats, where one group received a high-calcium diet (12 g Ca/kg diet) and the other a normal calcium diet (5 g Ca/kg diet) for 30 days, found no significant difference in cortical bone thickness between the two groups. This aligns with the outcomes of our experiment [13]. In summary, the intervention of ACC in our experiment had no significant impact on cortical bone thickness.

## 5. Conclusions

ACC supplementation can enhance osteoblast metabolism and inhibit osteoclast metabolism, resulting in a higher bone formation rate compared to bone resorption. This led to increased trabecular bone thickness and higher BMD.

## Figures and Tables

**Figure 1 biology-13-00201-f001:**
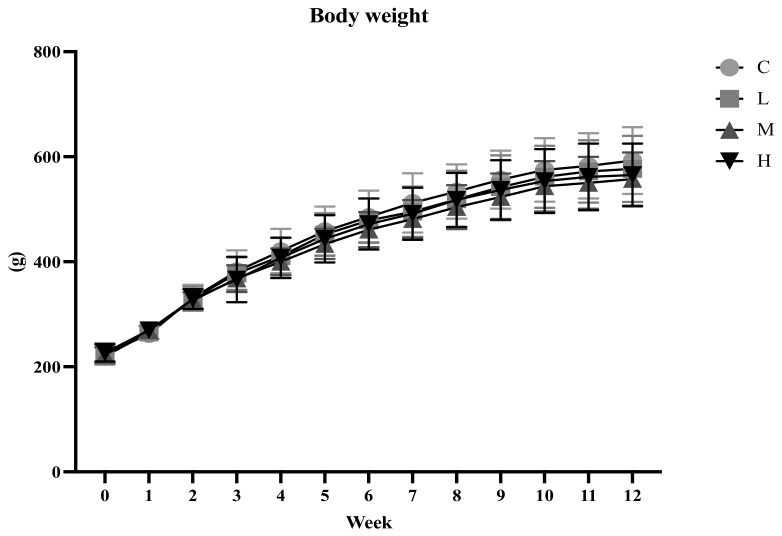
Body weights from weeks 0 to 12. All values are the mean ± standard deviation. There were no differences within the group by one-way ANOVA followed by Duncan’s multiple-range test. Abbreviations: C, control group; L, low-dose group (20.65 mg amorphous calcium carbonate (ACC)/kg body weight (BW)); M, medium-dose group (206.5 mg ACC/kg BW); H, high-dose group (413 mg ACC/kg BW).

**Figure 2 biology-13-00201-f002:**
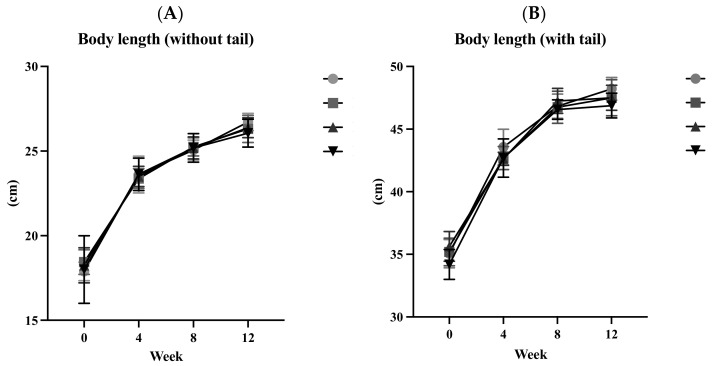
Body lengths from weeks 0 to 12. (**A**) Without tail; (**B**) with tail. All values are the mean ± standard deviation. There were no differences within the groups from the one-way ANOVA followed by Duncan’s multiple-range test. Abbreviations: C, control group; L, low-dose group (20.65 mg amorphous calcium carbonate (ACC)/kg body weight (BW)); M, medium-dose group (206.5 mg ACC/kg BW); and H, high-dose group (413 mg ACC/kg BW).

**Figure 3 biology-13-00201-f003:**
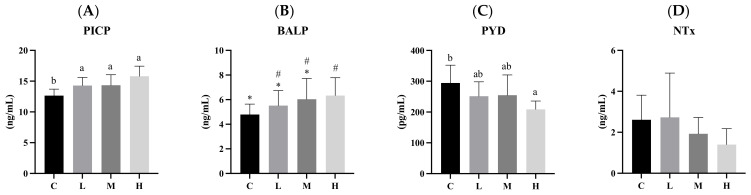
Serum bone turnover biomarker (BTM) concentrations after 12 weeks of the amorphous calcium carbonate (ACC) intervention. (**A**) Procollagen-I C-terminal propeptide (PICP), (**B**) bone alkaline phosphatase (BALP), (**C**) pyridinoline (PYD), and (**D**) cross-linked N-telopeptide (NTx). All values are the mean ± standard deviation. Different letters (a, b) indicate a significant difference between different groups at *p* < 0.05 from the one-way ANOVA with Duncan’s multiple-range test. Different symbols (#, *) indicate a significant difference between different groups at *p* < 0.1 from the one-way ANOVA with Duncan’s multiple-range test. Abbreviations: C, control group; L, low-dose group (20.65 mg amorphous calcium carbonate (ACC)/kg body weight (BW)); M, medium-dose group (206.5 mg ACC/kg BW); and H, high-dose group (413 mg ACC/kg BW).

**Figure 4 biology-13-00201-f004:**
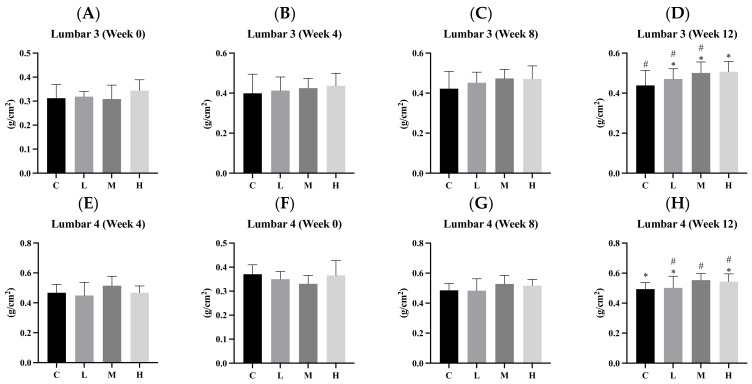
Bone mineral density after 0, 4, 8 and 12 weeks of the amorphous calcium carbonate (ACC) intervention. (**A**–**D**) Lumbar 3, (**E**–**H**) lumbar 4, and (**I**–**L**) left femur. All values are the mean ± standard deviation. Different letters (a, b, c) indicate a significant difference between different groups at *p* < 0.05 from the one-way ANOVA with Duncan’s multiple-range test. Different symbols (#, *) indicate a significant difference between different groups at *p* < 0.1 from one-way ANOVA with Duncan’s multiple-range test. Abbreviations: C, control group; L, low-dose group (20.65 mg amorphous calcium carbonate (ACC)/kg body weight (BW)); M, medium-dose group (206.5 mg ACC/kg BW); and H, high-dose group (413 mg ACC/kg BW).

**Figure 5 biology-13-00201-f005:**
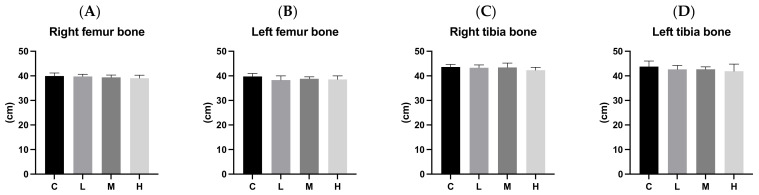
Bone lengths after 12 weeks of the amorphous calcium carbonate (ACC) intervention. (**A**) Right femur bone lengths; (**B**) left femur bone lengths; (**C**) right tibia bone lengths; and (**D**) left tibia bone lengths. All values are the mean ± standard deviation. There were no differences within groups from the one-way ANOVA followed by Duncan’s multiple-range test. Abbreviations: C, control group; L, low-dose group (20.65 mg amorphous calcium carbonate (ACC)/kg body weight (BW)); M, medium-dose group (206.5 mg ACC/kg BW); and H, high-dose group (413 mg ACC/kg BW).

**Figure 6 biology-13-00201-f006:**
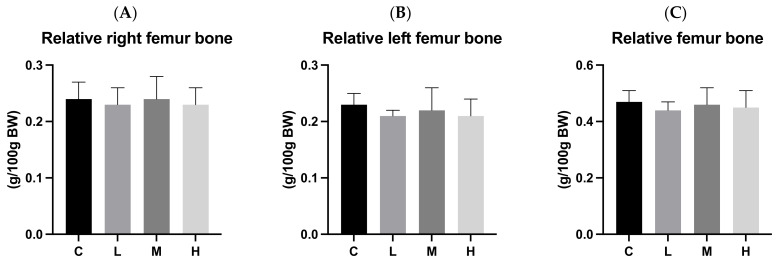
Bone weights after 12 weeks of the amorphous calcium carbonate (ACC) intervention. (**A**) Relative right femur bone weights; (**B**) relative left femur bone weights; (**C**) relative femur bone weights; (**D**) relative right tibia bone weights; (**E**) relative left tibia bone weights; and (**F**) relative tibia bone weights. All values are the mean ± standard deviation. There were no differences within groups from the one-way ANOVA followed by Duncan’s multiple-range test. Abbreviations: C, control group; L, low-dose group (20.65 mg amorphous calcium carbonate (ACC)/kg body weight (BW)); M, medium-dose group (206.5 mg ACC/kg BW); and H, high-dose group (413 mg ACC/kg BW).

**Figure 7 biology-13-00201-f007:**
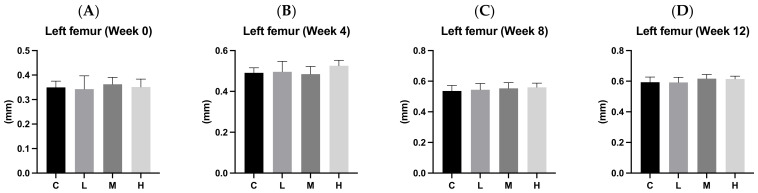
Left femur cortical bone thicknesses after 12 weeks of the amorphous calcium carbonate (ACC) intervention. (**A**) Week 0; (**B**) week 4; (**C**) week 8; and (**D**) week 12. All values are the mean ± standard deviation. There were no differences within groups from the one-way ANOVA followed by Duncan’s multiple-range test. Abbreviations: C, control group; L, low-dose group (20.65 mg amorphous calcium carbonate (ACC)/kg body weight (BW)); M, medium-dose group (206.5 mg ACC/kg BW); and H, high-dose group (413 mg ACC/kg BW).

**Figure 8 biology-13-00201-f008:**
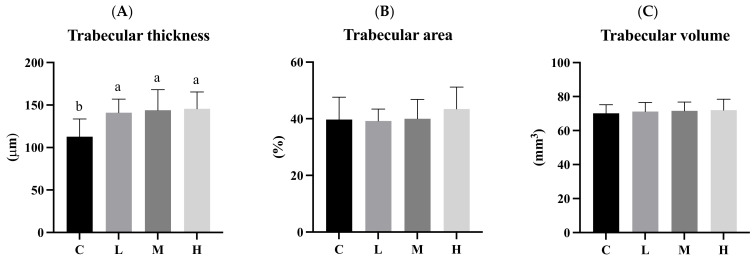
Left femur histomorphometrically after 12 weeks of the amorphous calcium carbonate (ACC) intervention. (**A**) Trabecular thickness; (**B**) trabecular area; and (**C**) trabecular volume. All values are the mean ± standard deviation. Different letters (a, b) indicate a significant difference between different groups at *p* < 0.05 from the one-way ANOVA with Duncan’s multiple-range test. Abbreviations: C, control group; L, low-dose group (20.65 mg amorphous calcium carbonate (ACC)/kg body weight (BW)); M, medium-dose group (206.5 mg ACC/kg BW); and H, high-dose group (413 mg ACC/kg BW).

**Figure 9 biology-13-00201-f009:**
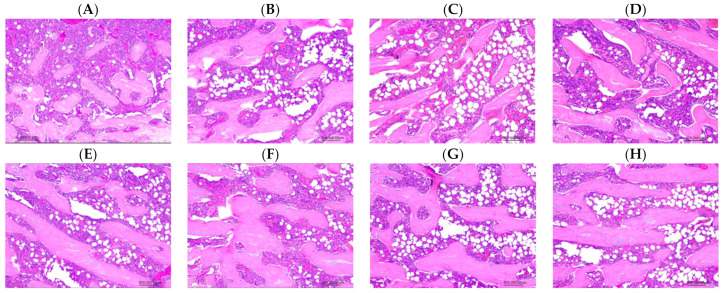
Representative images of histopathologic findings (50×, H&E). (**A**) Animal no.: C-1, (**B**) animal no.: C-5, (**C**) animal no.: L-12, (**D**) animal no.: L-14, (**E**) animal no.: M-19, (**F**) animal no.: M-22, (**G**) animal no.: H-27, and (**H**) animal no.: H-33. Abbreviations: C, control group; L, low-dose group (20.65 mg amorphous calcium carbonate (ACC)/kg body weight (BW)); M, medium-dose group (206.5 mg ACC/kg BW); and H, high-dose group (413 mg ACC/kg BW).

**Table 1 biology-13-00201-t001:** Body weight gain, body length gain, and food intake.

	C (*n* = 9)	L (*n* = 8)	M (*n* = 8)	H (*n* = 8)
Initial BW (g)	224.83 ± 20.50	220.19 ± 15.78	222.88 ± 15.54	226.63 ± 16.46
Total BW gain (g)	367.83 ± 45.44	356.69 ± 52.62	334.50 ± 50.66	338.81 ± 50.39
Total BL gain (with tail) (g)	13.17 ± 0.61	12.31 ± 1.13	11.88 ± 0.74	12.69 ± 1.22
Total BL gain (without tail) (g)	8.83 ± 0.50	7.88 ± 0.83	8.13 ± 1.03	8.06 ± 1.64
Food intake (g/day/rat)	25.67 ± 2.91	25.99 ± 0.58	24.76 ± 1.32	24.60 ± 2.33
Food efficiency (%)	14.33 ± 0.80	13.72 ± 1.98	13.47 ± 1.53	13.78 ± 1.54

All values are the mean ± standard deviation. Food efficiency was calculated by applying the following equation: food efficiency = (total body weight gain (g)/food intake (g)). There were no differences among the groups from the one-way ANOVA followed by Duncan’s multiple-range test. Abbreviations: BW, body weight; BL, body length; C, control group; L, low-dose group (20.65 mg amorphous calcium carbonate (ACC)/kg BW); M, medium-dose group (206.5 mg ACC/kg BW); and H, high-dose group (413 mg ACC/kg BW).

**Table 2 biology-13-00201-t002:** Lee’s index (LI).

	C (*n* = 9)	L (*n* = 8)	M (*n* = 8)	H (*n* = 8)
Week 0 (g/cm)	0.340 ± 0.01	0.328 ± 0.02	0.334 ± 0.03	0.341 ± 0.03
Week 1 (g/cm)	0.333 ± 0.01	0.324 ± 0.02	0.325 ± 0.01	0.319 ± 0.08
Week 2 (g/cm)	0.322 ± 0.02	0.328 ± 0.03	0.319 ± 0.01.	0.318 ± 0.01.
Week 3 (g/cm)	0.323 ± 0.01	0.318 ± 0.01	0.315 ± 0.01	0.318 ± 0.02
Week 4 (g/cm)	0.318 ± 0.01	0.316 ± 0.01	0.314 ± 0.01	0.313 ± 0.01
Week 5 (g/cm)	0.316 ± 0.01	0.313 ± 0.01	0.311 ± 0.00	0.318 ± 0.01
Week 6 (g/cm)	0.326 ± 0.01	0.321 ± 0.01	0.323 ± 0.01	0.329 ± 0.01
Week 7 (g/cm)	0.326 ± 0.01	0.328 ± 0.01	0.323 ± 0.01	0.319 ± 0.01
Week 8 (g/cm)	0.324 ± 0.01	0.319 ± 0.01	0.316 ± 0.01	0.319 ± 0.01
Week 9 (g/cm)	0.316 ± 0.01	0.316 ± 0.01	0.319 ± 0.01	0.321 ± 0.01
Week 10 (g/cm)	0.313 ± 0.01	0.318 ± 0.02	0.315 ± 0.01	0.320 ± 0.01
Week 11 (g/cm)	0.316 ± 0.01 ^ab^	0.318 ± 0.01 ^b^	0.306 ± 0.01 ^ab^	0.309 ± 0.01 ^a^
Week 12 (g/cm)	0.313 ± 0.01	0.318 ± 0.00	0.313 ± 0.01	0.318 ± 0.01

All values are the mean ± standard deviation. Different letters (a, b) indicate a significant difference between different groups at *p* < 0.05 from the one-way ANOVA with Duncan’s multiple-range test. Abbreviations: C, control group; L, low-dose group (20.65 mg amorphous calcium carbonate (ACC)/kg body weight (BW)); M, medium-dose group (206.5 mg ACC/kg BW); and H, high-dose group (413 mg ACC/kg BW). Lee’s index = cube root of body weight (g)/nose-to-anus length (cm).

## Data Availability

The original contributions presented in the study are included in the article, further inquiries can be directed to the corresponding authors.

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
