# Peer review of "Amorphous Calcium Carbonate from Plants Can Promote Bone Growth in Growing Rats"

_biology, 2024, doi:10.3390/biology13030201_

Round 1

Reviewer 1 Report

Comments and Suggestions for Authors

Scientific Manuscript ID: biology-2886373

The manuscript has been well formulated and addresses important gaps regarding bone metabolism. It encompasses various analyses with significant scientific impact. In order to improve the interpretation of the approaches, the authors should address the following points:

Line 27 – Reconfigure the Keywords, replacing items that are part of the title.

Line 137 – In the statistical analysis, specify the normality test used.

Line 152 – Remove Figure 1.

Line 167 – Remove Figure 2.

Line 234 – Reformulate the presentation of Table 8.

Line 260 – Prepare a panel for Figure 5.

Line 399 – The conclusions, although of great importance, do not encompass the proposed objectives.

Additional Comments:

  1. 1. The paper's main question is regarding Bone metabolism, with comparative modeling based on adjuvant administration. 2.The results are positive and fill a specific gap related to bone metabolism. The possibility of replication and practical application of bone modifications. Clearly applicable to different animal models. 3. This paper adds Recent data employing different evaluation methods, Points of relevance and impact compared to related literature. Especially regarding the model used. 4. No measures need modification. Specifically, only mentioning the normality test used in the statistical analysis. No further controls need to be considered.. 5.The conclusions extrapolate the presented objectives. In summary, those related to application. Otherwise, only adjustments will be necessary to reconcile objectives and conclusions. 6. The references are appropriate.

Author Response

Thanks for your excellent review and comments. We have incorporated the necessary changes in the revised manuscript point by point based on your comments. We have highlighted the changes in the original manuscript by using the red-color text.
Comment 1:
The manuscript has been well formulated and addresses important gaps regarding bone
metabolism. It encompasses various analyses with significant scientific impact. In order to
improve the interpretation of the approaches, the authors should address the following points:
Line 27 – Reconfigure the Keywords, replacing items that are part of the title.
Line 137 – In the statistical analysis, specify the normality test used.
Line 152 – Remove Figure 1.
Line 167 – Remove Figure 2.
Line 234 – Reformulate the presentation of Table 8.
Line 260 – Prepare a panel for Figure 5.
Line 399 – The conclusions, although of great importance, do not encompass the proposed
objectives.
Response :
Thank you for your comments. We have reconfigured the keywords, replacing items that
are part of the title (in line 38). Additionally, we have incorporated the use of normal
distribution tests into the statistical analysis in the Materials and Methods section (in line 151-
152). To enhance the quality of data presentation, we have utilized GraphPad Prism version 9.0
for corrections in Figures 1, 2, 5, and Table 8. Furthermore, we have modified the proposed
objectives from the conclusion (in line 479-480).

Reviewer 2 Report

Comments and Suggestions for Authors

In this work, Chun-Kai etc. studied the effect of amorphous calcium carbonate (ACC) supplementation on the bone growth in growing rats. Specifically, four groups of rats (control group, low-dose, medium-dose, and high-dose) were treated with different dosage of ACC for 12 weeks. The rats’ body length (BL) and weight (BW), Lee’s index, bone mineral density (BMD), bone turnover biochemical markers (BTMs), and bone length and weight were monitored and analyzed. The statistical analysis results show that ACC supplementation has no significant impact on the BL, BM, as well as bone length and weight. Besides that, the authors found that the group with calcium supplementation show increased trabecular bone thickness than the control group. In addition, the BTMs study results indicate that high dosage of ACC promoted the bone synthesis compared to the low dosage. The introduction, experimental design, and results discussion are complete. Here are my comments:

1. In this study, the ACC supplementation has no significant impact on most parameters including the body and bone length & weight. Will a higher dosage of calcium (i.e., 1000mg/kg/day) give a different conclusion? How about studying rats at different weeks? Can the author comment on that?   

2. Scale bar is missing in Figure 1 & 2.

3. Minor: Page 10, line 216, there is a typo: “ after 0, 4, 8 and 12 weeks of the a …”

Author Response

Thanks for your excellent review and comments. We have incorporated the necessary changes in the revised manuscript point by point based on your comments. We have highlighted the changes in the original manuscript by using the red-color text.
l Major Comments:
Comment 1:
In this study, the ACC supplementation has no significant impact on most parameters
including the body and bone length & weight. Will a higher dosage of calcium (i.e.,
1000mg/kg/day) give a different conclusion? How about studying rats at different weeks? Can
the author comment on that?
Response:
Previous studies have indicated that three-week-old rats were provided with either a low calcium diet (2.5 g Ca/kg diet) or a high-calcium diet (10 g Ca/kg diet) daily through ad libitum feeding or adjusted feeding until they reached 40 weeks of age. It was observed that the rate of
weight gain gradually slowed down after 15 weeks of age. The study found that the rats in the high-calcium diet group had significantly higher body weights compared to those in the low calcium
group at 4 and 13 weeks. However, the author suggested that this difference may be attributed to the higher food intake in the high-calcium diet group. Conversely, no significant difference in body weight was observed in the control diet group. This aligns with the outcomes
of our experiment. Furthermore, after increasing the duration of feeding, calcium intake did not affect the body weight of rats. In the ad libitum fed group, the bone length of the high calcium group was significantly greater than that of the low-calcium group at 4 and 13 weeks. However, in the adjusted fed group, there was no significant difference observed. The authors hypothesize that heavier rats tend to have longer bone lengths. Nonetheless, when there was no significant difference in body weight, there was also no significant difference in bone length between the rats given the high-calcium diet and those given the low-calcium diet. This aligns with the outcomes of our experiment. Furthermore, after increasing the duration of feeding,
calcium intake did not affect the bone length of rat [1]. Otherwise, during the growth and development stage, the body is particularly sensitive to factors that influence future health, such as BMD and peak bone mass (PBM). As a result, bone length and weight are not as
significantly affected as BMD [2]. Previous studies showed that a calcium deficiency did not impact bone length in growing rats [3]. Similarly, providing growing rats with either a normal calcium diet (5 g Ca/kg diet) or a high-calcium diet (12 g Ca/kg diet) for 30 days did not affect their height or bone length [4]. During the growth and development process, variations in calcium intake have less impacts on bone length and weight but are more likely to affect BMD and BTM.
In our study, body length was relatively steep from 0 to 4 weeks and gradually slowed down from 4 to 12 weeks. Therefore, we speculate that extending the duration of intervention will not significantly impact body length. Previous studies indicate that adequate calcium intake is limited to supporting normal bone growth and development, and it will not cause an individual to grow taller than their predetermined genetic height. Height is primarily determined by genetic factors but can also be influenced by hormones, environmental factors, and nutrition. Therefore, while calcium is crucial for bone health and growth, it is not the sole factor determining a person’s height [5-8]. Currently, there are few studies on higher calcium doses (e.g., 1000mg/kg/day). However,
for the dosage used in our study, body length, body weight, bone length, and bone weight did not exhibit an increasing trend with the dose escalation. It is inferred that the lack of observable
effects may be attributed to the dosage not being sufficiently high. Future studies could investigate whether higher calcium doses affect body length, body weight, bone length, and bone weight. Additionally, excessive intake of calcium supplements may lead to side effects,
such as interference with the body's absorption of iron and zinc, which must be taken into consideration.
Comment 2:
Scale bar is missing in Figure 1 & 2.
Response:
Thank you for your reminder, we have revised it in Figure 1 and 2.
l Minor Comments:
Comment 1:
Page 10, line 216, there is a typo: “ after 0, 4, 8 and 12 weeks of the a …”
Response:
Thank you for your reminder, we have revised in line 250 of the revised manuscript.

Reviewer 3 Report

Comments and Suggestions for Authors

The study demonstrates the effect of ACC intervention in growing rats as a potential therapeutic intervention for osteoporosis.  

In this article while the results and conclusions drawn are interesting, the data presentation requires significant improvements. A lot of the data is presented in tables but plots are highly encouraged to convey the importance of the findings.  Please use software like Prism/GraphPad if feasible, to improve quality of data presentation.  

Major

1. Authors should explain why Ntx as a measure of rate of bone loss was not found to be significant in Ca-treated mice relative to control. They state that it could be because this was performed in growing mice.  However, the authors should discuss the limitation of this study in terms of treating osteoporosis which is primarily a disorder that affects an aging patient population.  Authors should mention follow up studies that are warranted in aged rodents that have been treated by ACC supplementation in their growth period.   

2. The tabulated values for Body weight, body length, Lee's Index, food intake, bone weight, cortical bone thickness are distracting in the flow of the manuscript and should be plotted instead of in a table.    

3. The quality of the plots must be improved. Please use graphpad or another data application to plot the figures.  Also any data that meets significance should be annotated with an asterix for clarity. Using 'cd' or 'de' is confusing to the reader.

Minor

1. In the introduction, authors should briefly state the advantages of ACC over CCC and calcium citrate instead of just citing the relevant literature.  

Author Response

Thanks for your excellent review and comments. We have incorporated the necessary changes in the revised manuscript point by point based on your comments. We have highlighted the changes in the original manuscript by using the red-color text.
l Major Comments:
Comment 1:
Authors should explain why Ntx as a measure of rate of bone loss was not found to be significant in Ca-treated mice relative to control. They state that it could be because this was performed in growing mice. However, the authors should discuss the limitation of this study
in terms of treating osteoporosis which is primarily a disorder that affects an aging patient population. Authors should mention follow up studies that are warranted in aged rodents that have been treated by ACC supplementation in their growth period.
Response:
Thank you for your comments. In our study, we use 3-week-old male Wistar rats tosimul ate childhood and adolescent growth stages. The results of our study indicate that although supplementing ACC during the growth period has positive effects on bone mineral density (BMD), bone turnover markers (BTMs) and trabecular thickness, no relevant follow-up studies were conducted on elderly rats. However, osteoporosis primarily affects the elderly population. From the existing results, we can only infer that supplementing ACC during the
growth period can improve bone health and potentially reduce the risk of osteoporosis in the future. Nonetheless, we cannot ascertain the impact on bone health in old age, thus this aspect is a limitation of our study. One study observed that ovariectomized rats aged 16-17 weeks
were administered the same calcium content of ACC, CCC, and calcium citrate for 90 days (approximately 13 weeks). The trabecular bone mineral density of the femur and lumbar vertebrae in the ACC group was significantly higher than that of the CCC group and the
calcium citrate group. Additionally, the bone resorption biomarker, Dihydropyrimidine (DPD), was significantly lower in the ACC group compared to the CCC group and the calcium citrate group. These findings suggest that ACC is more effective than the two most common calcium supplements (CCC and calcium citrate) in preventing bone loss [9]. To build upon the positive findings regarding skeletal health, and previous study has demonstrated the excellent performance of ACC in preventing bone loss. In the future, the duration of feeding could be
extended, and follow-up studies could be conducted on elderly rodents treated with ACC supplements during the growth period. It is anticipated that, compared with the control group not receiving calcium supplements, they will exhibit better bone status (e.g.,BMD), thereby potentially reducing the incidence of osteoporosis in the elderly. (in line 431-453)
Comment 2:
The tabulated values for Body weight, body length, Lee's Index, food intake, bone weight, and cortical bone thickness are distracting in the flow of the manuscript and should be plotted instead of in a table.
Response:
Thank you for your suggestion. We have employed GraphPad Prism version 9.0 to improve the quality of data presentation and have utilized figures instead of tables (in Figure 1-9).
Comment 3:
The quality of the plots must be improved. Please use graphpad or another dataapplication to plot the figures. Also, any data that meets significance should be annotated with an asterix for clarity. Using 'cd' or 'de' is confusing to the reader.
Response:
Thank you for your suggestion. In footnotes, we have used ’# and ✽’ instead of ‘c, d and e’ (In Figure 3 and 4).
l Minor Comments:
Comment 1:
In the introduction, authors should briefly state the advantages of ACC over CCC and calcium citrate instead of just citing the relevant literature.
Response:
Thank you for your comments. We have added the information in the introduction to briefly state the advantages of ACC over CCC and calcium citrate. (in line 71-73)
References
1. Bollen, A.-M.; Bai, X.-Q. Effects of long-term calcium intake on body weight, body fat and
bone in growing rats. Osteoporosis international 2005, 16, 1864-1870.
2. Sengupta, P. The laboratory rat: relating its age with human's. International journal of
preventive medicine 2013, 4, 624.
3. Hernández-Becerra, E.; Jímenez-Mendoza, D.; Mutis-Gonzalez, N.; Pineda-Gomez, P.;
Rojas-Molina, I.; Rodríguez-García, M.E. Calcium Deficiency in Diet Decreases the
Magnesium Content in Bone and Affects Femur Physicochemical Properties in Growing
Rats. Biol Trace Elem Res 2020, 197, 224-232, doi:10.1007/s12011-019-01989-9.
4. Viguet-Carrin, S.; Hoppler, M.; Membrez Scalfo, F.; Vuichoud, J.; Vigo, M.; Offord, E.A.;
Ammann, P. Peak bone strength is influenced by calcium intake in growing rats. Bone 2014,
68, 85-91, doi:10.1016/j.bone.2014.07.029.
5. Weaver, C.M.; Peacock, M. Calcium. Adv Nutr 2011, 2, 290-292, doi:10.3945/an.111.000463.
6. Baird, G.S. Ionized calcium. Clin Chim Acta 2011, 412, 696-701, doi:10.1016/j.cca.2011.01.004.
7. Straub, D.A. Calcium supplementation in clinical practice: a review of forms, doses, and
indications. Nutr Clin Pract 2007, 22, 286-296, doi:10.1177/0115426507022003286.
8. Winzenberg, T.; Shaw, K.; Fryer, J.; Jones, G. Calcium supplements in healthy children do
not affect weight gain, height, or body composition. Obesity 2007, 15, 1789-1798.
9. Shaltiel, G.; Bar-David, E.; Meiron, O.E.; Waltman, E.; Shechter, A.; Aflalo, E.D.; Stepensky,
D.; Berman, A.; Martin, B.R.; Weaver, C.M. Bone loss prevention in ovariectomized rats
using stable amorphous calcium carbonate. 2013.